molecular biology/cellular biology/neuroscience

oligodendrocyte, differentiation, myelin, multiple sclerosis, *Dendropanax morbiferus*

**Author for correspondence:**
Hyun-Jeong Yang
e-mail: yang@ube.ac.kr

# *Dendropanax morbiferus* leaf extract facilitates oligodendrocyte development

Ji-Young Kim[1], Ju-Young Yoon[2], Yuki Sugiura[3], Soo-Kyoung Lee[4], Jae-Don Park[5], Gyun-Jee Song[6] and Hyun-Jeong Yang[2,7]

[1]Department of Anesthesiology and Pain Medicine, Yonsei University College of Medicine, Seoul 120-749, Republic of Korea
[2]Department of Integrative Biosciences, University of Brain Education, Cheonan 31228, Republic of Korea
[3]Department of Biochemistry and Integrative Medical Biology, School of Medicine, Keio University, Tokyo 160-8582, Japan
[4]Department of Health Science and Daily Sports, Global Cyber University, Cheonan 31228, Republic of Korea
[5]Cheju Halla University, Jeju 63092, Republic of Korea
[6]Department of Medical Science, International St Mary's Hospital, Catholic Kwandong University, Gangneung, Republic of Korea
[7]Korea Institute of Brain Science, Seoul, Republic of Korea

H-JY, 0000-0002-5262-6721

Treatment of multiple sclerosis is effective when anti-inflammatory, neuroprotective and regenerative strategies are combined. *Dendropanax morbiferus* (*DM*) has anti-inflammatory, anti-oxidative properties, which may be beneficial for multiple sclerosis. However, there have been no reports on the effects of *DM* on myelination, which is critical for regenerative processes. To know whether *DM* benefits myelination, we checked differentiation and myelination of oligodendrocytes (OLs) in various primary culture systems treated with *DM* leaf EtOH extracts or control. *DM* extracts increased the OL membrane size in the mixed glial and pure OL precursor cell (OPC) cultures and changed OL-lineage gene expression patterns in the OPC cultures. Western blot analysis of *DM*-treated OPC cultures showed upregulation of MBP and phosphorylation of ERK1/2. In myelinating cocultures, *DM* extracts enhanced OL differentiation, followed by increased axonal contacts and myelin gene upregulations such as Myrf, CNP and PLP. Phytochemical analysis by LC-MS/MS identified multiple components from *DM* extracts, containing bioactive molecules such as quercetin, cannabidiol, etc. Our results suggest *DM* extracts enhance OL differentiation, followed by an increase in membrane size and axonal contacts, thereby indicating enhanced myelination. In addition, we found that *DM*

extracts contain multiple bioactive components, warranting further studies in relation to finding effective components for enhancing myelination.

# 1. Introduction

*Dendropanax morbiferus* (*DM*) grows in eastern Asia, central and south America and the Malaysian peninsula [1]. Recent scientific researches on *DM* have shown its antioxidant effects [2,3], protective effects on human dopaminergic cells [4], induction of human leukaemia U937 cell apoptosis [5] and of human osteosarcoma cell autophagy [6], anti-thrombotic effects [7] and anti-inflammatory [8], particularly neuroinflammatory, effects [9].

Multiple sclerosis is a major demyelinating disease of the central nervous system leading to focal plaque of primary demyelination and diffuse neurodegeneration in the grey and white matter of the brain and spinal cord [10].

Patients with multiple sclerosis show increased oxidative stress markers and inflammation [11]. Recent studies on *DM* suggest that *DM* might be effective on multiple sclerosis symptoms because of its antioxidant [2,3], cell-protective [4] and anti-inflammatory properties [8,9].

Remyelination fails in many chronically demyelinated multiple sclerosis plaques [12,13]. The remyelination failure is mainly attributable to reduced oligodendrocyte precursor cell (OPC) recruitment and differentiation [14]. Among these events, it is thought that disrupted differentiation is the major time-limiting factor because OPC mitogen and recruitment factor PDGF overexpression do not increase remyelination [15]. Currently, it is understood that oligodendrocyte (OL) differentiation failure is the biggest cause of remyelination failure [16]. Study of various developmental markers of OLs has shown that OPC is not fully differentiated in multiple sclerosis lesions [12,13,17,18]. Therefore, enhancement of OL differentiation and myelination is an important problem to solve in relation to multiple sclerosis treatment.

Although recent research has reported that *DM* has anti-oxidative, cell-protective and anti-inflammatory properties, which can be useful in multiple sclerosis treatment, the link between *DM*'s benefits and myelin-forming OLs has not been investigated yet. In the present study, we aimed to reveal the effects of *DM* on OL differentiation, and myelination using multiple primary culture systems, including *in vitro* myelination cultures.

# 2. Material and Methods

## 2.1. Mice

Timed pregnant females (day 13.5 of pregnancy) or pups (postnatal day 0–2) of CrljOri:CD1(ICR) mouse line were purchased from ORIENT BIO Inc. (Seongnam, Korea). All experiments were performed in compliance with the relevant laws and institutional guidelines and were approved by the University of Brain Education's Animal Care and Use Committee (approval no. 2017-AE-01).

## 2.2. Preparation of plant extracts

The leaves, stem barks and roots of 6-, 20- and 50-year-old *DM* were collected on Jeju island and were identified and provided by Mago-Yeongnong-Johap (Jeju, Korea). Other dried plants, except *DM*, were identified and purchased from Jirisan Cheongjung Yakcho (Sancheong, Kyoungnam, Korea): *Eucommia ulmoides* (*EU*), *Achyranthes aspera* (*AcA*), *Xanthium strumarium* (*XS*), *Astragalus membranaceus* (*AM*), *Artemisia princeps* (*ArP*), *Actinidia polygama* (*AP*), *Rehmannia glutinosa* (*RG*), *Ledebouriella seseloides* (*LS*) and *Artemisia annua* (*AA*). The product year of these plants was 2016. All the specimens were deposited in the University of Brain Education Herbarium. The deposition numbers are as follows: 6-year-old *DM* stem (2017-05-3), leaf (2017-12-1) and root (2017-12-2); 20-year-old *DM* leaf (2017-13); 50-year-old *DM* stem (2017-05-2) and leaf (2017-14); *EU* (2017-03); *AcA* (2017-07); *XS* (2017-08); *AM* (2017-09); *ArP* (2017-04); *AP* (2017-01); *RG* (2017-06); *LS* (2017-11) and *AA* (2017-10). Dried plant samples were chopped and ground into a fine powder because lowering particle size increases surface-contact area between samples and extraction solvents, resulting in efficient extraction [19]. Eight grams of each powder were soaked into EtOH with up to 50 ml volume. Plant samples were

incubated in the shaking rotator for one week at room temperature. After one week, the insoluble materials were removed by centrifugation and the resulting supernatant was collected and concentrated by evaporation at 70°C. The extracts were adjusted to $8\,g\,ml^{-1}$ and stored in aliquots at $-20°C$ [19,20].

## 2.3. Cell cultures

Mixed glial cultures were prepared from cortices of mice at postnatal days 0–2 on poly-D-lysine (PDL)-coated flasks or coverslips and were maintained in DMEM/F-12 medium containing 10% fetal bovine serum, 5% horse serum and 1× penicillin–streptomycin (Gibco). For OPC pure cultures, OPCs were isolated by shaking from the flasks containing mixed glial cultures on days in vitro (DIV) 10. After removing the astrocytes by dish-panning, the OPCs were seeded on PDL-coated coverslips and were maintained in Sato medium (DMEM containing 1× B-27 supplement, 1× Glutamax, 1× penicillin–streptomycin, 1% horse serum, 1× sodium pyruvate, $0.34\,\mu g\,ml^{-1}$ T3 and $0.4\,\mu g\,ml^{-1}$ T4). For cocultures, mouse dorsal root ganglia (DRG) neuronal cultures and mouse glial mixed cultures were separately prepared in advance. DRG neurons were prepared as described previously [21]. The OPCs were isolated by shaking and were seeded on DRG neuronal cultures and were maintained in coculture medium (DMEM containing B-27 supplement, N-2 supplement, $5\,\mu g\,ml^{-1}$ N-Acetyl-cysteine, $5\,\mu M$ forskolin and penicillin–streptomycin). For plant extract treatment, the above prepared plant extracts were diluted with the medium into 1 : 1000 (high, if not indicated) or 1 : 100 000 (low) and filtered through a $0.22\,\mu m$ syringe filter. Cells were incubated with the plant extract-diluting medium or control medium for DIV 8–12 in the mixed glial cultures, for DIV 1–3 in the pure OPC cultures and for DIV 1–7 or 1–12 for DRG/OPC cocultures. Control or plant extract-containing medium was changed with fresh medium every two days. If not indicated, the leaf part of six-year-old DM was used in the treatment.

## 2.4. Immunofluorescence

For immunocytochemistry, the cells were fixed with 4% paraformaldehyde (PFA)/phosphate-buffered saline (PBS) for 15 min at room temperature; washed with PBS; and blocked for 45 min at room temperature with blocking solution (PBS containing 5% normal donkey serum, 0.5% Triton X-100 and 0.05% sodium azide). The samples were incubated overnight at 4°C with primary antibodies diluted in blocking solution, washed three times in PBS, incubated for 45 min with secondary antibodies and washed in PBS and mounted with mounting medium (Vectashield H-1400, vectorlabs). For primary antibodies, the following antibodies were used: rat monoclonal antibody to MBP (1 : 300, MAB386, Chemicon), mouse hybridoma supernatants to O4 (1 : 5), rabbit polyclonal antibodies to Olig2 (1 : 500, AB9610, Chemicon) and Caspr (1 : 1000) [22]. Secondary antibodies were obtained from Jackson Immunoresearch. DAPI was obtained from Invitrogen.

## 2.5. Real-time PCR

Total RNA was isolated with TRIzol (Sigma) from cultured OPC cells on DIV 3 or myelinating cultures on DIV12. The isolated RNA was treated with DNaseI to eliminate genomic DNA before reverse transcription. Mouse cDNAs were prepared using SuperScript II Reverse Transcriptase (Invitrogen). Specific primer sets were used for Olig1 (forward, 5′-GCTCGCCCAGGTGTTTTGT-3′; reverse, 5′-GCATGGAACGTGGTTGGAAT-3′), Id2 (forward, 5′-CCTGCATCACCAGAGACCTG-3′; reverse, 5′-TTCGACATAAGCTCAGAAGGGAA-3′), Ascl1 (forward, 5′-CAACCGGGTCAAGTTGGTCA-3′; reverse, 5′-CTCATCTTCTTGTTGGCCGC-3′), MBP (forward, 5′-CCAGAGCGGCTGTCTCTTCC-3′; reverse, 5′-CATCCTTGACTCCATCGGGCGC-3′), Myrf (forward, 5′-TGGCAACTTCACCTACCACA-3′; reverse, 5′-GTGGAACCTCTGCAAAAAGC-3′), CNP (forward, 5′-GTTCTGAGACCCTCCGAAAA-3′; reverse, 5′-CCTTGGGTTCATCTCCAGAA-3′), PLP (forward, 5′-GGTACAGAAAAGCTAATTGAGA CC-3′; reverse, 5′-GATGACATACTGGAAAGCATGA-3′) and GAPDH (forward, 5′-GGTCGGTGTGAA CGGATTTG-3′; reverse, 5′-TCGTTGATGGCAACAATCTCCACT-3′). Real-time PCR was performed in the Step One Plus Real-Time PCR System (Applied Biosystems) using SYBR Green mix. All reactions were performed in triplicate and GAPDH was used for normalization.

## 2.6. Western blot analysis

Western blot analysis was performed using OPC cultures lysed in loading buffer, and chemiluminescence was detected using Amersham Imager 600 (GE Healthcare Life Sciences). Rat antibody against MBP (1 : 500, MAB386, Chemicon), rabbit antibodies against phospho-p42/44 (1 : 1000, 4370P, Cell signaling), and β-actin (1 : 5000, Abcam) were used for immunoblotting.

## 2.7. Image analysis

Images were obtained using a Leica TCS SPE microscope and Axio Imager Z1 equipped with Apotome (Carl Zeiss). Image analysis was performed using Image J. The images were given to the investigators without the sample names but only with numbers for analysis. Thus, the images were scored blinded to the plant names. For MBP$^+$ or Caspr$^+$ area per cell, the images were manually analysed in a random manner to detect single cells under the same threshold throughout all images. If distinguishing a single cell was difficult because of the density, the cells were excluded from the analysis. For Olig2$^+$ or Dapi$^+$ cell number, the cells were automatically detected and counted under the same threshold throughout all images in Image J. For O4$^+$ or MBP$^+$ area per field, marker-positive areas were automatically detected and measured under the same threshold throughout all images in Image J.

## 2.8. Statistical analyses

All graph data are presented as the mean ± s.e.m. Statistical analyses were performed using unpaired Student's $t$-test with two tails, unequal variance. Sample size was based on similar studies in the field.

## 2.9. Non-targeted metabolome analysis

*DM* leaf ethanol extracts were stored at −80°C until analyses and phytochemicals were analysed by liquid chromatography (LC)-mass spectrometry (MS)/MS. For non-targeted analysis, MS (Q-Exactive focus, Thermo Fisher Scientific, San Jose, CA, USA), which enables us to perform highly selective and sensitive metabolite quantification owing to the Fourier transfer MS principle, was connected to a high-performance liquid chromatography (Ultimate3000 system, Thermo Fisher Scientific). The samples were resolved on Acculaim C18 (2.1 mm ID × 150 mm, 3 μm particle, Thermo Fisher Scientific), using a step gradient with mobile phase A (60 : 40 (v/v) of water: acetonitrile in 10 mM ammonium formate and 0.1% formic acid) and mobile phase B (90 : 10 (v/v) of isopropyl alcohol: acetonitrile in 10 mM ammonium formate and 0.1% formic acid) at ratios of 68 : 32 (0−4 min), 55 : 45 (4−5 min), 48 : 52 (5−8 min), 34 : 66 (8−11 min), 30 : 70 (11−14 min) and 25 : 75 (14−18 min), at a flow rate of 0.2 ml min$^{-1}$ and a column temperature of 45°C. The Q-Exactive focus mass spectrometer was operated under an ESI-positive mode for all detections. Full mass scan ($m/z$ 50−900), followed by three rapid data-dependent MS/MS, was operated at a resolution of 70 000. The automatic gain control target was set at $3 \times 10^6$ ions, and the maximum ion injection time was 100 ms. Source ionization parameters were optimized with the spray voltage at 3 kV and the other parameters were as follows: transfer temperature at 320°C, S-Lens level at 50, heater temperature at 300°C, sheath gas at 36 and auxilliary gas at 10.

Compound Discoverer 2.0 (Thermo Fisher Scientific) was used for the non-targeted metabolomics workflow as described in Zhou [23]. Briefly, this software first aligned the total ion chromatograms of different samples along the retention time. Then the detected features were extracted and merged into components. The resulting compounds were identified by both (i) formula prediction based on accurate $m/z$ value and isotope peak patterns, and (ii) MS/MS structural validation querying to $m/z$ cloud database (https://www.mzcloud.org/). Moreover, formula predicted signals were assigned into candidate compounds by database search (ChemSpider database; http://www.chemspider.com/).

# 3. Results

## 3.1. *Dendropanax morbifera* extracts increased the oligodendrocyte membrane size in the mixed glial cultures

To check whether *DM* extract conferred any effects on the formation of the OL membrane sheath, we performed mixed glial cultures from the cortices of the postnatal days 0−2 ICR pups on PDL-coated

coverslips. The cultures were initially incubated in the regular glial medium, and after eight days the medium was changed to Sato medium containing either the ethanol control (figure 1$a$,$c$) or the following medicinal plant extracts, which have been appeared as their neurological recovery function in traditional literature [24]: $DM$ (figure 1$b$,$d$), $EU$, $AcA$, $XS$, $AM$, $ArP$, $AP$, $RG$, $LS$ and $AA$ (figure 1$a$−$e$; electronic supplementary material, figure S1). On DIV 12, the cultures were fixed and stained with MBP to determine the size of the OL membrane sheath. MBP is the major marker of mature myelinating OLs [21]; therefore, we used MBP-positive percentage area per field on DIV 12 as the indication of membrane formation. Using Image J, MBP-positive percentage area per field was measured. The pictures were taken in low magnification to include as many mature OLs as possible. Many of the investigated medicinal plant extracts in the study including $DM$ leaf extract induced increases in MBP-positive membrane size compared with EtOH control (figure 1$e$). $ArP$ and $DM$ showed the highest effects on MBP-positive percentage area per field in the glial mixed culture system. The glial mixed cultures contain three major glial cell types: OLs, astrocytes and microglia. Therefore, the effects of $DM$ on OL membrane sheath formation may derive not only from the OLs but also from the combined effects of the three cell types, because astrocytes and microglia also affect myelin synthesis [25,26].

## 3.2. Dendropanax morbiferus extracts increased the size of oligodendrocyte membrane sheath in the isolated oligodendrocyte precursor cell cultures

To clarify whether the changes in membrane size by $DM$ extracts derive directly from OL or indirectly through astrocytes or microglia, we isolated OPCs from the mixed glial culture on DIV 10 by shaking. Isolated OPCs were seeded on PDL-coated coverslips and incubated in Sato medium containing differentiating hormones T3 and T4 [21]. On DIV 1, the medium was changed to fresh medium containing either the control EtOH (figure 1$f$,$l$) or the following plant extracts: $DM$ (figure 1$g$,$m$), $AA$ (figure 1$h$), $AP$ (figure 1$i$), $EU$ (figure 1$j$), $LS$ (figure 1$k$), $DF$ and $ArP$ (figure 1$n$). The cultures were fixed on DIV 4 and were then used for MBP staining. The images were taken and blindly analysed using Image J. MBP staining images were converted into 8 bit pictures (electronic supplementary material, figure S2 h), and all single cells per field were identified manually and numbered (electronic supplementary material, figure S2i). MBP-positive area per cell was calculated using the Image J (figure 1$n$). In pure OPC cultures, $DM$ extract-treated OLs showed the largest membrane size (figure 1$n$), suggesting that the $DM$ extracts contain components that directly enhance OL membrane synthesis independent of other glial cell types. Unlike the results in the glia mixed culture, $ArP$ showed only half of the membrane size compared with $DM$-treated OLs in pure OPC cultures, suggesting its action may require the combinatory mechanism with astrocytes or microglia.

## 3.3. Dendropanax morbiferus showed different effects on oligodendrocyte membrane synthesis depending on its part

$DM$ may have different biological activities depending on its parts [27,28]. To clarify the part of $DM$ that would be the most effective for OL membrane synthesis, we tested leaves, stems and roots of $DM$. Pure OPC cultures were prepared and control EtOH or $DM$ extracts were added from DIV 1, and the cultures were fixed on DIV 3. Compared with control EtOH, $DM$ leaf extracts induced a marked increase in MBP-positive area per cell (***$p < 0.001$, $n = 3$ experiments, Student's $t$-test), $DM$ roots showed a relatively small increase (*$p < 0.05$, $n = 3$ experiments, Student's $t$-test), and $DM$ stem did not show any significant changes (figure 1$o$). MBP-positive areas per cell were dependent on the concentration of the extracts (electronic supplementary material, figure S2j). A higher concentration of $DM$ leaf led to a larger OL membrane size than lower concentration. This suggests that the effective components of $DM$ on OL membrane synthesis exist strongly in the leaves of $DM$.

## 3.4. Dendropanax morbiferus leaf extracts induced changes in gene expressions related with OL differentiation

To check whether $DM$ leaf altered the gene expression in OLs, we used pure OPC cultures and treated them with control EtOH or $DM$ leaf extract of low or high concentrations (figure 2$a$−$d$, see Material and Methods for concentrations) from DIV 1 and collected cells to isolate RNA at DIV 3, followed by cDNA

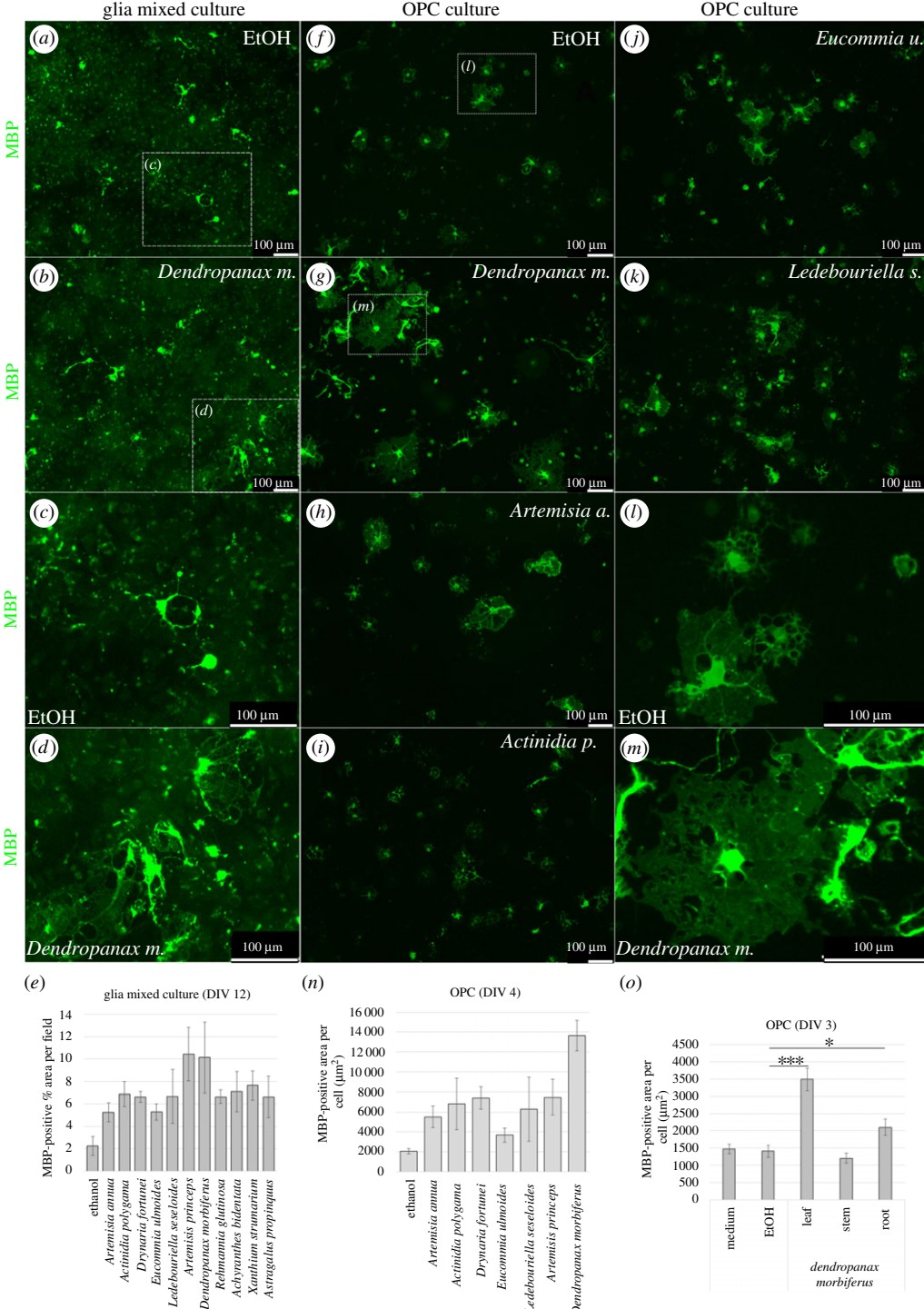

**Figure 1.** *Dendropanax morbiferus* leaf EtOH extract treatment increased the size of OL membrane sheath in the mixed glial culture as well as pure OL culture. (*a–e*) Primary mixed glial cultures were treated with the medium containing EtOH or several medicinal plant EtOH extracts during DIV 8–12 and fixed at DIV 12 for MBP staining to analyse the membrane size. Compared with EtOH-treated cultures (*a,c*), the size of OL membrane sheath was increased by the treatment of medicinal plant extracts including *DM* (*b,d*; electronic supplementary material, figure S1). (*e*) Total summary of MBP-positive percentage area per field depending on the treatment. Among the investigated medicinal plants, *ArP* and *DM* induced the highest MBP-positive percentage area per field in the glia mixed culture. (*f–n*) OPCs were isolated from the mixed glial cultures on DIV 10 and seeded on PDL-coated coverslip. On DIV 1, the OPCs were treated with the EtOH- or the indicated medicinal plant extract-containing medium and fixed on DIV 4 for immunostaining with MBP antibody. *Dendropanax morbiferus*-treated OPCs showed the highest MBP-positive area per cell, while *ArP*-treated OPCs indicated the half of the membrane size of *DM*-treated OPCs. (*o*) MBP-positive area per cell depending on the indicated plant parts of *DM*. The leaf showed the highest effects on the increment of the OL membrane size. Bars represent mean ± s.e.m. *$p < 0.05$; ***$p < 0.001$; $n = 3$ experiments, Student's *t*-test. Scale bars, 100 μm (*a–d,f–m*).

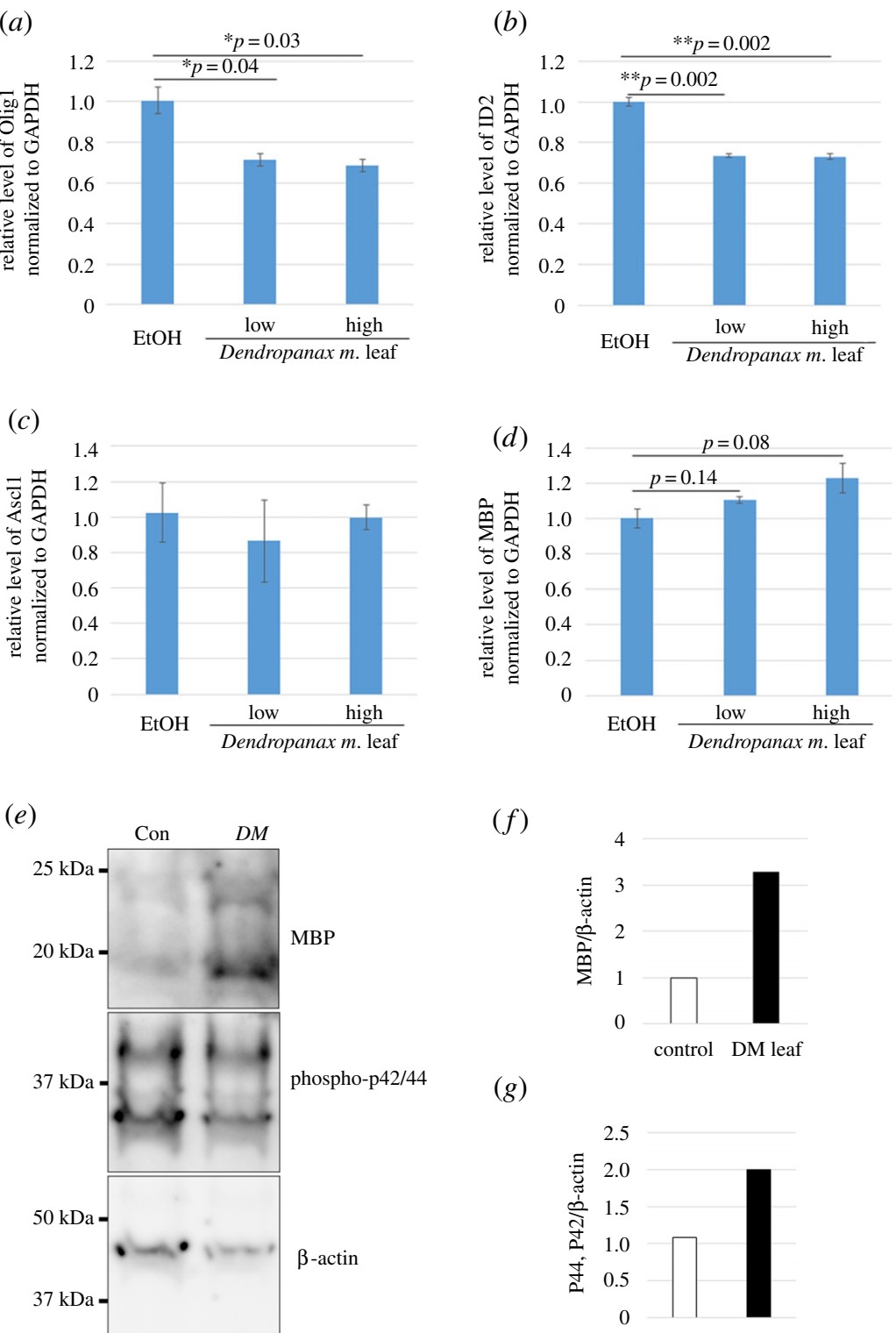

**Figure 2.** *Dendropanax morbiferus* leaf extracts downregulated inhibitory factors and upregulated promoting factors for OL differentiation in pure OPC cultures. (*a*–*d*) OPCs were treated with EtOH- or *DM* leaf extract-containing medium during DIV 1–3 and were used for RNA extraction and cDNA synthesis, followed by real-time PCR to ascertain the relative expression pattern of the following genes: (*a*) *Olig1*, (*b*) *ID2*, (*c*) *Ascl1* and (*d*) *MBP*. The expression of *Olig1* and *ID2* but not *Ascl1* was significantly downregulated, while the expression of *MBP* was in a tendency to increase by *DM* leaf extract treatment. Bars represent mean $\pm$ s.e.m. *$p < 0.05$; **$p < 0.01$; $n = 3$ experiments, Student's *t*-test. (*e*) Three different cultures were pooled and used for western blot analysis to detect MBP and phospho-p42/44. β-actin was used for normalization. Compared with EtOH-treated cultures, upregulations in MBP (*f*) and in phosphorylation of p42/44 (*g*) were observed in the *DM* leaf extract-treated cultures.

synthesis and real-time PCR. Gene expressions related to OL differentiation, such as *Olig1*, *ID2*, *Ascl1* and *MBP*, were investigated. *GAPDH* was used to normalize the expression of the indicated genes. The relative expression of *Olig1* and *ID2* were significantly decreased (figure 2*a*,*b*), while that of MBP showed a tendency of increase (figure 2*d*) compared with EtOH control. The relative expression of Ascl1 was not changed by the treatment (figure 2*c*). During OPC differentiation, the expressions of *Olig1* [29], *ID2* [30] and *Ascl1* [31] are downregulated, whereas *MBP* expression is upregulated [32]. Therefore, real-time PCR results suggest that the *DM* leaf extract contains biologically effective components for OL differentiation, supporting the immunocytochemistry data, which showed the positive effects of *DM* leaf on OL membrane synthesis.

Next, to see whether we can find changes in protein levels, we performed western blot analysis in the same system on DIV 3 using OPCs treated with either *DM* leaf or control during DIV 1–3. Consistent with the real-time PCR results, MBP expression normalized by β-actin was upregulated in *DM* leaf extract-treated OPC cultures compared with the control-treated OPC cultures (figure 2*e*,*f*; electronic supplementary material, figure S3). The extracellular signal-regulated kinase (ERK) signalling pathway plays a specific role in the timing of OL differentiation [33]. Therefore, we also checked the phosphorylation status of ERK1/2 to see whether the altered differentiation by *DM* leaf extract affects ERK signalling. The results showed that the *DM* leaf extract treatment increased the phosphorylation of ERK1/2 compared with the control treatment (figure 2*e*,*g*), suggesting that the enhanced OL differentiation by *DM* leaf extract is mediated by ERK signalling to a certain extent.

## 3.5. *Dendropanax morbiferus* leaf extracts enhanced the oligodendrocyte differentiation in the myelinating cultures

Increases in OL membrane size (figure 1) and changes in OL gene expression (figure 2) in OPC cultures by *DM* leaf extracts suggest that *DM* leaf extracts may enhance OL differentiation in OPC cultures. *In vivo*, there are numerous neuronal signals affecting OL differentiation and myelin synthesis. Therefore, to know whether the *DM* leaf extracts affect OL differentiation even under the existence of neurons, we used *in vitro* myelinating cultures, in which pure OPCs are seeded on DRG neurons. To analyse the OL differentiation step, we used DIV 7 cocultures for immunostaining of antibodies against Olig2 (positive for all OL stages), O4 (positive from premature OL stage) and MBP (positive from mature OL stage) (figure 3*a*). At this time point (DIV 7), myelin is not formed; however, OL differentiation can be distinguished by the indicated markers of various OL developmental stages. For *in vitro* cocultures, *DM* leaf extracts of different age were used to check whether or not they show the same tendency in their effects on OL membrane development. The number of Olig2-positive cells per field (figure 3*c*) as well as O4-positive area per cell (figure 3*d*) remained unchanged. The MBP-positive area per cell significantly increased in cocultures incubated with 6- and 20-year-old *DM* leaf extracts (*$p < 0.05$, Student's *t*-test, $n = 3$ experiments, figure 3*e*) but not with 50-year-old *DM* leaf extract (figure 3*e*). These results suggest that extract from *DM* leaf enhances the OL differentiation not only in pure OPC cultures (figure 1*f*–*o*) but also in DRG/OPC cocultures (figure 3). Moreover, it indicates that *DM* leaf does not have a significant effect on OPC proliferation (figure 3*c*).

## 3.6. *Dendropanax morbiferus* leaf extract increased the axonal contacts of oligodendrocytes in the myelinating cultures

We found that *DM* leaf extract enhanced OL differentiation on DIV 7 in DRG/OPC cocultures (figure 3). Next, we asked whether the facilitated differentiation enhances axonal contacts of OLs to initiate myelination. To test this, we immunostained cocultures on DIV 12 with antibodies against Caspr, which clusters upon OL contact and ensheathment [34,35]. Caspr-positive area per cell was measured and compared between EtOH- (figure 4*a*) and *DM* leaf extract-treated cultures (figure 4*b*). The result showed that *DM* leaf extract significantly facilitated Caspr clustering in DRG/OPC coculture system on DIV 12 (*$p < 0.05$, Student's *t*-test, $n = 3$ experiments, figure 4*d*). This suggests that *DM* leaf extract is effective not only for OL differentiation but also in boosting axonal contacts.

To confirm the immunocytochemical analysis in the myelinating culture system, we performed quantitative real-time PCR on myelin genes by using the same culture system on DIV12. Myelin genes such as Olig1, Myrf, CNP, MBP and PLP were investigated. Myrf is a transcription factor activating the expression of myelin genes [36]. CNP is a non-compact myelin protein [37]; MBP and PLP are important structural components of compact myelin [38]. We found that all of the investigated myelin

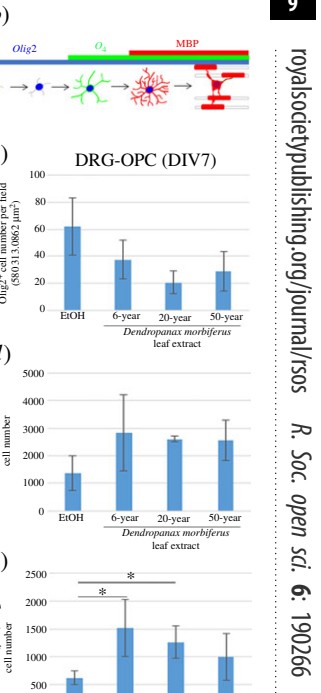

**Figure 3.** *Dendropanax morbiferus* leaf extract enhanced the OL differentiation in the late stage of OL development in the myelinating cultures. (*a*) For *in vitro* cocultures, OPCs were seeded on dorsal root ganglion neuronal cultures (DIV 0) and incubated with the medium containing control or *DM* leaf extract of the indicated ages. Fresh medium was added every two days. On DIV 7, cocultures were incubated with O4-containing medium and subsequently fixed for further staining with antibodies to MBP, Olig2 and Dapi. (*b*) OL developmental markers used in the analysis are indicated. Antibodies to Olig2 (blue), O4 (green) and MBP (red) stain OL-lineage cells from precursor, immature and mature stages, respectively. Image analyses were performed in the following aspects: (*c*) Olig2-positive cell number per field, (*d*) O4-positive area per an OL-lineage cell and (*e*) MBP-positive area per an OL-lineage cell. There were no significant changes in the early stage of the OL development by *DM* leaf treatment (*c,d*). However, *DM* leaf of 6- and 20-year-old but not 50-year-old significantly increased MBP-positive area per cell at DIV 7 (*e*). Bars represent mean ± s.e.m. *$p < 0.05$; $n = 3$ experiments, Student's *t*-test. Scale bar, 100 μm.

genes increased its expression by *DM* leaf treatment and significant increases were observed in Myrf, CNP and PLP (*$p < 0.05$, **$p < 0.01$, Student's *t*-test, $n = 3$ experiments, figure 4*e*). The increase in myelin gene expression is consistent with the immunocytochemical analysis (figure 4*a*–*d*), supporting the facilitating effects of DM leaf on myelination.

## 3.7. Chemical composition of *Dendropanax morbiferus* leaf extract by LC-MS/MS analysis

To know the chemical constituents of *DM*, we performed LC-MS/MS analysis by using *DM* leaf EtOH extracts. A total of 300 chemicals were assigned (data not shown) and their molecular weights were matched with databases. Among these, compounds which have been reported as their bioactivities are listed in table 1. The lists include quercetin [39–41], chlorogenic acid [42], rutin [39,40], carnosol [43], dextromethorphan [44], cannabidiol [45], bremazocine [46], doxapram [47], resolvin D2 [48], procyclidine [49–51], 2-arachidonoylglycerol [52–55] and eplerenone [56].

# 4. Discussion

In this paper, we report the function of *DM* on the development of OL-lineage cells: from OPC proliferation till myelination. OPCs proliferate to supply OLs producing enough myelin for the nervous system. By *DM* treatment, the number of OLs was not significantly changed (figure 3*c*), suggesting *DM* does not affect the proliferation of OPCs. In the OPC stage, they express

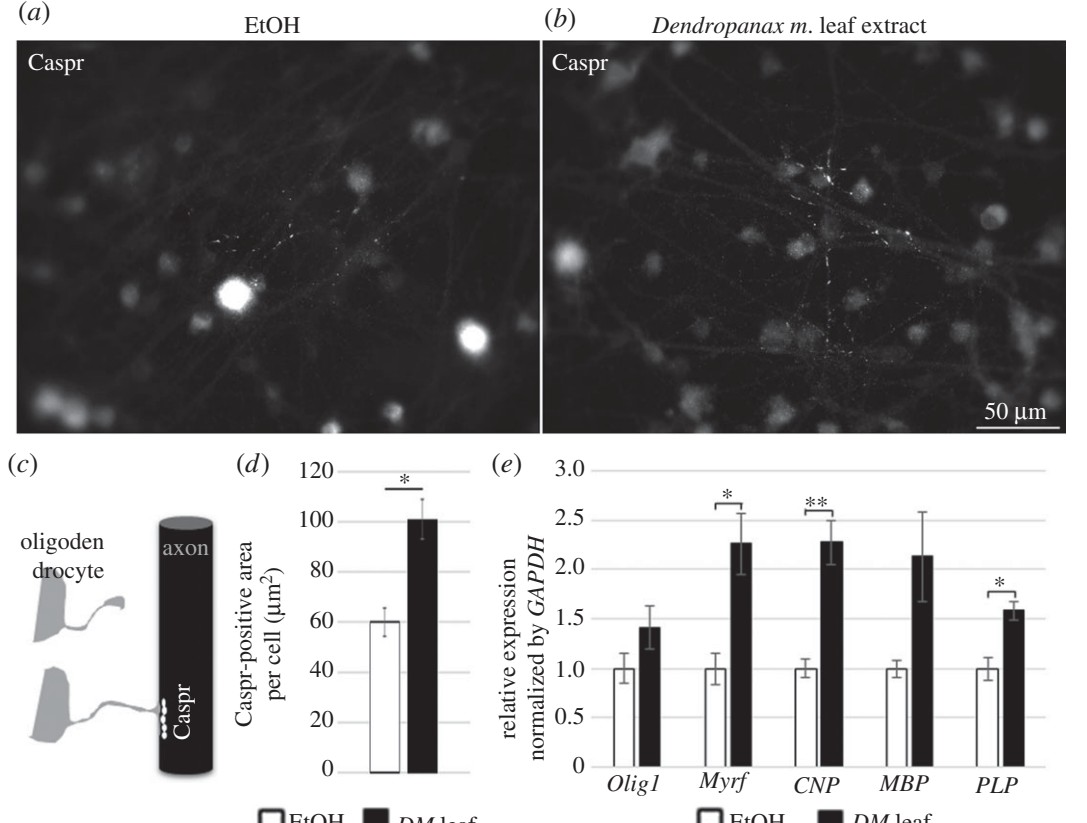

**Figure 4.** *Dendropanax morbiferus* leaf extract increased the axonal contacts of OLs on DIV 12 in the myelinating cultures. (*a–d*) DRG–OPC cocultures were incubated with either EtOH (*a*) or *DM* leaf EtOH extract (*b*) containing medium during DIV 1–12 with regular medium changes every two days. On DIV 12, cocultures were fixed for Caspr staining to compare the axonal contacts. (*c*) A drawing description showing axonal Caspr clustering by OL contacts. (*d*) Image analysis of Caspr staining. All Caspr-positive area per cell was combined and compared between EtOH- and *Dendropanax morbiferus* leaf extract-treated cocultures. Bars represent mean ± s.e.m. *$p < 0.05$; $n = 3$ experiments, Student's *t*-test. Scale bar, 50 μm. (*e*) Quantitative real time PCR on the myelinating cultures treated with EtOH- or *DM* leaf extract during DIV 1–12. Several important myelin genes such as *Myrf*, *CNP* and *PLP* showed a significant increase in their expression by *DM* leaf extract treatment in the myelinating culture system.

differentiation inhibitory transcription factors such as ID2 [57]. *DM* treatment reduced ID2 expression (figure 2*b*), suggesting its facilitation of OL differentiation. Indeed, the expression of MYRF, a transcription factor of myelinating OL [57], was significantly increased by *DM* treatment in myelinating cultures on DIV12 (figure 4*e*), supporting this notion. Consistently, myelin genes such as CNP and PLP showed a significant increase in their expression in *DM*-treated myelinating cultures (figure 4*e*). MBP protein expression was also increased in all of the investigated culture systems by *DM* treatment (figures 1, 2*e* and 3). This suggests that *DM* facilitates the OL differentiation process from OPC to mature OLs forming myelin. The OPC differentiation is mediated by multiple combinatory mechanisms [58]. Among them, ERK signalling is one of the critical pathways in OL differentiation [33] as well as myelin growth [59]. Our results indicate that *DM* leaf extract enhances OL differentiation at least partially by activating ERK signalling pathway (figure 2*e,g*).

For myelination, the processes of OL should contact axons, and then they initiate ensheathment to form myelin by increasing their membrane size. To analyse the effects of DM on myelination steps, we checked the axonal contacts of OLs by using myelinating cultures with Caspr staining (figure 4), and investigated OL membrane synthesis by using glia mixed cultures, OPC cultures and myelinating cultures with MBP staining or expression analysis (figures 1–4). For immunocytochemical membrane synthesis analysis, we used mixed glial cultures and OPC cultures because they present membrane sheath in a flat round form without neurons, which makes analysis easier. *DM* treatment increased OL axonal contacts as well as OL membrane size compared with control treatment. This is consistent with the result of facilitated OL differentiation by *DM*.

**Table 1.** Chemical components of leaf ethanol extracts of *DM*. Phytochemicals of *DM* leaf ethanol extracts were investigated by LC-MS/MS analysis. Chemicals with reported bioactivities are listed.

| formula | suggested compound | suggested structure | MS/MS (database search score) | molecular weight | retention time (min) | peak area (arbitary unit) |
|---|---|---|---|---|---|---|
| C15 H10 O7 | quercetin |  | 89.7 | 302.04141 | 2.353 | 90 013.43876 |
| C16 H18 O9 | chlorogenic acid |  | 89.4 | 354.09412 | 2.043 | 411 599.9664 |
| C27 H30 O16 | rutin |  | 87.7 | 610.15157 | 2.365 | 754 591.1196 |
| C20H26O4 | carnosol |  | 75 | 330.18224 | 7.537 | 3 896 034.619 |
| C18 H25 N O | dextromethorphan |  | 66.9 | 271.19308 | 9.725 | 310 359.8127 |
| C21 H30 O2 | cannabidiol |  | — | 314.22411 | 9.727 | 38 541.99824 |
| C20 H29 N O2 | (—)-bremazocine |  | — | 315.21991 | 8.211 | 3223.952411 |
| C24 H30 N2 O2 | doxapram |  | — | 378.2296 | 10.403 | 3724.664755 |

(*Continued.*)

**Table 1.** (*Continued.*)

| formula | suggested compound | suggested structure | MS/MS (database search score) | molecular weight | retention time (min) | peak area (arbitrary unit) |
|---|---|---|---|---|---|---|
| C22 H32 O5 | resolvin D2 |  | — | 376.22452 | 9.894 | 2921.592155 |
| C19 H29 N O | procylidine |  | — | 287.22368 | 1.595 | 696.0637427 |
| C23 H38 O4 | 2-arachidonoylglycerol |  | — | 378.27529 | 15.155 | 5438.022477 |
| C24 H30 O6 | eplerenone |  | — | 414.20274 | 4.293 | 4399.070776 |

Previous reports about *DM* are about antioxidant effects [2,3], cell-protective effects [4], selective cell death [5,6], anti-thrombotic [7] and anti-inflammatory effects [8,9]. Our observation of *DM* functions in enhancement in OL development is novel, raising possibilities of its potential use in multiple sclerosis treatment. Non-targeted metabolome analysis by LC-high-resolution MS assigned multiple compounds in DM ethanol extracts, showing DM is a complex of mixture of phytochemicals (table 1). They contain molecules having bioactivities, such as quercetin, chlorogenic acid, rutin, carnosol, dextromethorphan, cannabidiol, bremazocine, doxapram, resolvin D2, procyclidine, 2-arachidonoylglycerol and eplerenone. Interestingly, quercetin, dextromethorphan and cannabidiol protect OL-lineage cells from the stress environments [41,44,45], while quercetin also improves myelination in the context of perinatal cerebral hypoxia ischaemia-induced brain injury by promoting the proliferation of OPCs [41]. Other components have been reported for various functions. Quercetin, chlorogenic acid and rutin mediate the anti-oxidation process [39,42]. Carnosol and resolvin D2 are related to anti-inflammatory functions [43,48]. Bremazocine is a κ-opioid agonist [60] and Doxapram is a respiratory stimulant [61]. Procyclidine is a synthetic anticholinergic agent [49] and 2-Arachidonoylglycerol is an endocannabinoid that activates the cannabinoid receptors CB1 and CB2 [62–64]. Eplerenone has been reported to reduce both the risk of death and the risk of hospitalization among patients with systolic heart failure and mild symptoms [65]. As other components (except quercetin) have not been reported in relation with myelination, they warrant further studies to screen the effective components on myelination.

Accumulative findings suggest that treatment of progressive multiple sclerosis should be a combinatory strategy of anti-inflammation, neuroprotection and regeneration [66]. The reported functions of phytochemicals of *DM* leaf on inflammation and cell protection as well as our current finding of *DM* leaf function in myelination indicate that *DM* leaf extract can be considered as a candidate for relieving the symptoms of progressive multiple sclerosis, even as a complex. Medicinal plants used in multiple sclerosis care report their benefits in relieving spasticity, pain, tremor and depression [67] and enhancing fatty acid metabolism and lymphocyte function [68,69]. However, to our knowledge, no study has explored the function of medicinal plants on OL differentiation/myelination, which is a critical step for remyelination in multiple sclerosis, and our result that *DM* leaf extracts affect OL development is the first observation as far as we know.

In this research, we have shown that *DM* enhances endogenous OL differentiation, followed by membrane synthesis, using various types of primary cultures such as glial mixed culture, pure OPC culture and myelinating cultures. However, *in vivo* is a much more complicated environment than *in vitro*. Therefore, confirmation of the *DM* effects in *in vivo* demyelination models such as cuprizone assay, lysolecithin mode, or experimental allergic encephalomyelitis may be necessary in the future. In phytochemical analysis, we identified bioactive components whose functions in myelination have not been reported yet, warranting further studies on the isolation of active components for OL differentiation and myelination from the current isolated candidates.

# 5. Conclusion

*DM* leaf EtOH extract enhanced the differentiation of OL and membrane formation, at least partially, by ERK signalling pathway. Moreover, *DM* leaf extract increased the axonal contacts of mature OLs, which is an initial step for myelination, as well as myelin gene expression. It also contained multiple bioactive molecules, warranting further studies on searching for active components on myelination. Our findings suggest that *DM* leaf extract may contain a novel therapeutic target in the treatment of progressive multiple sclerosis.

Ethics. Permission to provide samples that were used in the study was granted to Mago-Yeongnong-Johap and Jirisan Cheongjung Yakcho by Gyungsangnamdo Sancheongguncheong (permit no. 2012-Gyungnamsancheong-00005). The study was reviewed and approved by the Institutional Animal Care and Use Committee (approval no. 2017-AE-01) of the University of Brain Education.
Data accessibility. All data have been submitted as figures in the main text or as electronic supplementary material.
Authors' contributions. J.-Y.K. and G.-J.S. participated in the design of the study and drafted the manuscript. J.-Y.Y. carried out the molecular laboratory work, participated in data analysis and carried out the statistical analyses. Y.S. carried out the mass spectrometry analysis. S.-K.L. and J.-D.P. coordinated the study. H.-J.Y. carried out the molecular laboratory work, participated in data analysis, participated in the design of the study and drafted the manuscript. All authors gave final approval for publication.
Competing interests. The authors declare no competing interests.

Funding. This work was supported by the Basic Science Research Program through the National Research Foundation of Korea (NRF) funded by the Ministry of Education (2017R1D1A3B03027875); University of Brain Education (2017-03) and a faculty research grant of Yonsei University College of Medicine for (6-2014-0045). The funder played no role in the design and conduct of the study; collection, management, analysis and interpretation of the data; preparation, review, or approval of the manuscript; and decision to submit the manuscript for publication.

Acknowledgements. We thank Dr Elior Peles for antibodies to O4 and Caspr.

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
