## [Reviewer comments · Royal Society Open Science]

Review History

RSOS-190266.R0 (Original submission)

Review form: Reviewer 1

Is the manuscript scientifically sound in its present form?

Yes

Are the interpretations and conclusions justified by the results?

Yes

Is the language acceptable?

Yes

Is it clear how to access all supporting data?

Yes

Do you have any ethical concerns with this paper?

No

Have you any concerns about statistical analyses in this paper?

No

Recommendation?

Accept with minor revision (please list in comments)

Comments to the Author(s)

Authors investigated the effects of *Dendropanax morbiferus* (DM) leaf extract on oligodendrocyte (OL) development in this study. DM leaf extract treatment facilitated OL differentiation and changed gene expression patterns in the OLs in both pure OPC cultures and myelinating cocultures. Furthermore, they analyzed the components of DM leaf extracts by LC-MS/MS, reporting novel components potentially effective on OL development.

This paper describes the effects of DM leaf extracts on OL development, which is a critical step for targeting multiple sclerosis, a disease having a broad spectrum of patients. As there is no cure for the multiple sclerosis currently regardless of high prevalence worldwide (2.3million people in 2015), novel targets appeared in this paper as well as basic science behind it have a value to attract broad audience of RSOS. I think this study is scientifically sound and useful to the community, therefore, relevant for the publication for RSOS in general.

The authors have originality on the report of basic biology about how DM leaf extracts affect OL development and on identification of the potential bioactive components.

The methods used in this research is convincing and relevant when judged by researches in the similar field. Authors used various primary cultures not the cell lines, which make the story stronger. Authors limited their analysis in oligodendrocytes only, which made the interpretation more clear. The interpretation of results are thought to be valid and appropriate.

Although this paper is thought to be suitable for the publication, followings are some minor points that authors can consider for the improvement of the paper.

1. page22, "P" (t-test) should be italic.
2. Authors used OL developmental markers in Figure 3. It will be nicer if they add additional image information, which readers can understand how the marker expression changes according to the developmental stage.
3. Same in Figure 4. To help understanding, graphic information about the marker will be helpful.

Review form: Reviewer 2**Is the manuscript scientifically sound in its present form?**

Yes

Are the interpretations and conclusions justified by the results?

Yes

Is the language acceptable?

Yes

Is it clear how to access all supporting data?

Yes

Do you have any ethical concerns with this paper?

No

Have you any concerns about statistical analyses in this paper?

No

Recommendation?

Accept as is

Comments to the Author(s)

Authors demonstrated that *Dendropanax morbiferus* (DM) extracts enhance oligodendrocyte differentiation, followed by increase in membrane size and axonal contacts, thereby indicating enhanced myelination. DM-treated OPC cultures showed upregulation of MBP and phosphorylation of ERK1/2 and enhanced myelin gene upregulations such as Myrf, CNP, and PLP. The manuscript was designed and described well. DM may be important or novel therapeutics for demyelinating disease. So this manuscript is suitable for publications.

Review form: Reviewer 3

Is the manuscript scientifically sound in its present form?

Yes

Are the interpretations and conclusions justified by the results?

Yes

Is the language acceptable?

Yes

Is it clear how to access all supporting data?

Yes

Do you have any ethical concerns with this paper?

No

Have you any concerns about statistical analyses in this paper?

I do not feel qualified to assess the statistics

Recommendation?

Accept with minor revision (please list in comments)

Comments to the Author(s)

The authors can show convincingly that the *dendropanax morbiferus* leaf EtOH extract increases oligodendrocyte differentiation and membrane sheath size in culture.

Comments:

- Figure1: In the figure legend, statistical tests are described, however in the figure there are no statistical tests in the figure.
- Figure1: The MBP-positive area per cell differs strongly between the OPC DIV4 14000 μm^2 (k) and OPC DIV3 3500 μm^2 (p). How do the authors explain this difference in the effect of

dendropanax morbiferus leaf extract on the MBP-area?

- Figure2: The Western Blot shows a strong increase in MBP signal with DM extract, but the beta-Actin surprisingly weak. In addition this experiment seems to be done only once. Would it be possible for the authors to show another replicate?

-Table 1 is added twice in the manuscript.

Decision letter (RSOS-190266.R0)

28-May-2019

Dear Dr Yang

On behalf of the Editors, I am pleased to inform you that your Manuscript RSOS-190266 entitled "Dendropanax Morbiferus Leaf Extract Facilitates Oligodendrocyte Development" has been accepted for publication in Royal Society Open Science subject to minor revision in accordance with the referee suggestions. Please find the referees' comments at the end of this email.

The reviewers and handling editors have recommended publication, but also suggest some minor revisions to your manuscript. Therefore, I invite you to respond to the comments and revise your manuscript.

- Ethics statement

- Data accessibility

<http://datadryad.org/submit?journalID=RSOS&manu=RSOS-190266>

- Competing interests

- Authors' contributions

All submissions, other than those with a single author, must include an Authors' Contributions section which individually lists the specific contribution of each author. The list of Authors should meet all of the following criteria; 1) substantial contributions to conception and design, or

acquisition of data, or analysis and interpretation of data; 2) drafting the article or revising it critically for important intellectual content; and 3) final approval of the version to be published.

- Acknowledgements

- Funding statement

Because the schedule for publication is very tight, it is a condition of publication that you submit the revised version of your manuscript before 06-Jun-2019. Please note that the revision deadline will expire at 00.00am on this date. If you do not think you will be able to meet this date please let me know immediately.

- 1) A text file of the manuscript (tex, txt, rtf, docx or doc), references, tables (including captions) and figure captions. Do not upload a PDF as your "Main Document";

- 2) A separate electronic file of each figure (EPS or print-quality PDF preferred (either format should be produced directly from original creation package), or original software format);
- 3) Included a 100 word media summary of your paper when requested at submission. Please ensure you have entered correct contact details (email, institution and telephone) in your user account;
- 4) Included the raw data to support the claims made in your paper. You can either include your data as electronic supplementary material or upload to a repository and include the relevant doi within your manuscript. Make sure it is clear in your data accessibility statement how the data can be accessed;
- 5) All supplementary materials accompanying an accepted article will be treated as in their final form. Note that the Royal Society will neither edit nor typeset supplementary material and it will be hosted as provided. Please ensure that the supplementary material includes the paper details where possible (authors, article title, journal name).

on behalf of Prof Catrin Pritchard (Subject Editor)
openscience@royalsociety.org

Associate Editor Comments to Author:

Three reviewers have provided commentary on your paper, and the general view is that the manuscript is on track for publication subject to a number of minor tweaks, which are outlined by the reviewer reports. Please ensure that you fully respond to and incorporate the recommended changes in your revision. Good luck!

Reviewer comments to Author:

Reviewer: 1

Comments to the Author(s)

Authors investigated the effects of *Dendropanax morbiferus* (DM) leaf extract on oligodendrocyte (OL) development in this study. DM leaf extract treatment facilitated OL differentiation and changed gene expression patterns in the OLs in both pure OPC cultures and myelinating cocultures. Furthermore, they analyzed the components of DM leaf extracts by LC-MS/MS, reporting novel components potentially effective on OL development.

This paper describes the effects of DM leaf extracts on OL development, which is a critical step for targeting multiple sclerosis, a disease having a broad spectrum of patients. As there is no cure for the multiple sclerosis currently regardless of high prevalence worldwide (2.3million people in 2015), novel targets appeared in this paper as well as basic science behind it have a value to attract broad audience of RSOS. I think this study is scientifically sound and useful to the community, therefore, relevant for the publication for RSOS in general.

The authors have originality on the report of basic biology about how DM leaf extracts affect OL development and on identification of the potential bioactive components.

The methods used in this research is convincing and relevant when judged by researches in the similar field. Authors used various primary cultures not the cell lines, which make the story stronger. Authors limited their analysis in oligodendrocytes only, which made the interpretation more clear. The interpretation of results are thought to be valid and appropriate.

Although this paper is thought to be suitable for the publication, followings are some minor points that authors can consider for the improvement of the paper.

1. page22, "P" (t-test) should be italic.
2. Authors used OL developmental markers in Figure 3. It will be nicer if they add additional image information, which readers can understand how the marker expression changes according to the developmental stage.
3. Same in Figure 4. To help understanding, graphic information about the marker will be helpful.

Reviewer: 2

Comments to the Author(s)

Authors demonstrated that *Dendropanax morbiferus* (DM) extracts enhance oligodendrocyte differentiation, followed by increase in membrane size and axonal contacts, thereby indicating enhanced myelination. DM-treated OPC cultures showed upregulation of MBP and phosphorylation of ERK1/2 and enhanced myelin gene upregulations such as Myrf, CNP, and PLP. The manuscript was designed and described well. DM may be important or novel therapeutics for demyelinating disease. So this manuscript is suitable for publications.

Reviewer: 3

Comments to the Author(s)

The authors can show convincingly that the *dendropanax morbiferus* leaf EtOH extract increases oligodendrocyte differentiation and membrane sheath size in culture.

Comments:

- Figure1: In the figure legend, statistical tests are described, however in the figure there are no statistical tests in the figure.
- Figure1: The MBP-positive area per cell differs strongly between the OPC DIV4 14000 μm^2 (k) and OPC DIV3 3500 μm^2 (p). How do the authors explain this difference in the effect of dendropanax morbiferus leaf extract on the MBP-area?
- Figure2: The Western Blot shows a strong increase in MBP signal with DM extract, but the beta-Actin surprisingly weak. In addition this experiment seems to be done only once. Would it be possible for the authors to show another replicate?
- Table 1 is added twice in the manuscript.

Author's Response to Decision Letter for (RSOS-190266.R0)

See Appendix A.

Decision letter (RSOS-190266.R1)

04-Jun-2019

Dear Dr Yang,

I am pleased to inform you that your manuscript entitled "Dendropanax Morbiferus Leaf Extract Facilitates Oligodendrocyte Development" is now accepted for publication in Royal Society Open Science.

on behalf of Mr Andrew Dunn (Associate Editor) and Catrin Pritchard (Subject Editor)
openscience@royalsociety.org

Associate Editor Comments to Author (Mr Andrew Dunn):

Associate Editor: 1

Comments to the Author:

(There are no comments.)

Reviewer comments to Author:

Appendix A

Reviewer comments to Author:

Reviewer: 1

Comments to the Author(s)

Authors investigated the effects of Dendropanax morbiferus (DM) leaf extract on oligodendrocyte (OL) development in this study. DM leaf extract treatment facilitated OL differentiation and changed gene expression patterns in the OLs in both pure OPC cultures and myelinating cocultures. Furthermore, they analyzed the components of DM leaf extracts by LC-MS/MS, reporting novel components potentially effective on OL development.

This paper describes the effects of DM leaf extracts on OL development, which is a critical step for targeting multiple sclerosis, a disease having a broad spectrum of patients. As there is no cure for the multiple sclerosis currently regardless of high prevalence worldwide (2.3million people in 2015), novel targets appeared in this paper as well as basic science behind it have a value to attract broad audience of RSOS. I think this study is scientifically sound and useful to the community, therefore, relevant for the publication for RSOS in general.

The authors have originality on the report of basic biology about how DM leaf extracts affect OL development and on identification of the potential bioactive components.

The methods used in this research is convincing and relevant when judged by researches in the similar field. Authors used various primary cultures not the cell lines, which make the story stronger. Authors limited their analysis in oligodendrocytes only, which made the interpretation more clear. The interpretation of results are thought to be valid and appropriate.

Although this paper is thought to be suitable for the publication, followings are some minor points that authors can consider for the improvement of the paper.

1. *page22, "P" (t-test) should be italic.*

=> In original manuscript, it is written in italic. During the process of converting the caption into PDF, the word format of the figure caption disappears. This should be corrected in the publication process. Thank you for the correction.

2. *Authors used OL developmental markers in Figure 3. It will be nicer if they add additional image information, which readers can understand how the marker expression changes according to the developmental stage.*

=> A graphic image of oligodendrocyte development with marker information is provided as new Figure 3(b). Therefore, labeling in the Figure 3 has been changed as follows: b, c, d (old Figure 3) into c, d, e (new Figure 3). Accordingly, following description about new Figure 3 is also included in the main text as follows:

"(Page17, Line 21-23) (b) Oligodendrocyte developmental markers used in the analysis are indicated. Antibodies to Olig2 (blue), O4 (green) and MBP (red) stain oligodendrocyte lineage cells from precursor, immature and mature stages, respectively."

3. *Same in Figure 4. To help understanding, graphic information about the marker will be helpful.*

=> A drawing description showing how the marker Caspr appears on axonal clustering is added in new Figure 4 (c). Therefore, labeling in the Figure 4 has been changed as follows: c, d (old Figure 4) into d, e (new Figure 4). Accordingly, following description about new Figure 4 is also included in the main text as follows:

“(Page17, Line 33-34) (c) A drawing description showing axonal Caspr clustering by oligodendrocyte contacts.”

Reviewer: 2

Comments to the Author(s)

Authors demonstrated that Dendropanax morbiferus (DM) extracts enhance oligodendrocyte differentiation, followed by increase in membrane size and axonal contacts, thereby indicating enhanced myelination. DM-treated OPC cultures showed upregulation of MBP and phosphorylation of ERK1/2 and enhanced myelin gene upregulations such as Myrf, CNP, and PLP. The manuscript was designed and described well. DM may be important or novel therapeutics for demyelinating disease. So this manuscript is suitable for publications.

Reviewer: 3

Comments to the Author(s)

The authors can show convincingly that the dendropanax morbiferus leaf EtOH extract increases oligodendrocyte differentiation and membrane sheath size in culture.

Comments:

- Figure1: In the figure legend, statistical tests are described, however in the figure there are no statistical tests in the figure.

=> Thank you very much for your correction. We corrected this in the new version of Figure 1.

- Figure1: The MBP-positive area per cell differs strongly between the OPC DIV4 14000 μm^2 (k) and OPC DIV3 3500 μm^2 (p). How do the authors explain this difference in the effect of dendropanax morbiferus leaf extract on the MBP-area?

=> Isolated OPC cultures contain oligodendrocyte lineage cells of mixed developmental stages. At DIV3, there are still many immature oligodendrocytes, however once oligodendrocytes enter into the mature stage, the size of membrane expands fast. Next figure (O'Meara et al., 2011) describes how fast the membrane expansion progresses after DIV3. The graph shows about 8-fold increase in MBP+ cell number and about 7.2-fold increase in MBP protein at DIV6 compared to DIV3. We measured MBP+ area and it was about less than 4-fold increase at DIV4 compared to DIV3 (new Figure 1n, o). As area is the most expressive factor in terms of oligodendrocyte maturity, our result is consistent with previous reports describing developmental stage of oligodendrocytes in vitro (O'Meara et al., 2011).

1) O'Meara RW et al. (2011) Derivation of enriched oligodendrocyte cultures and oligodendrocyte/neuron myelinating co-cultures from post-natal murine tissues. *J. Vis. Exp.* (54), e3324

- Figure2: The Western Blot shows a strong increase in MBP signal with DM extract, but the beta-Actin surprisingly weak. In addition this experiment seems to be done only once. Would it be possible for the authors to show another replicate?

=> The MBP antibody was stripped from the membrane and it was reblotted with beta-Actin antibody to normalize MBP signal by the exact loading amount of the cells. Stripping procedure may affect the signal of beta-Actin.

Figure 2e western blot is the result of pooled samples from three different cultures as written in the figure legend.

As reviewer3 requested, we performed another sets of pool experiments and another replicate was added in Supplementary Figure 3 (b-d). Consistent results were obtained from the experiments. According to this, explanation was added as follows:

“(Page 8, Line 20) electronic supplementary material, figure s3”

“(Page 20, Line 24-29) **Supplementary material figure S3. Increased expression of MBP in the oligodendrocyte precursor cell cultures by incubation with the *Dendropanax morbiferus* extract.** (a-b) For one set of western blot sample, three different oligodendrocyte precursor cell cultures were pooled at days in vitro 3 and used for western blot analysis to detect MBP and β -Actin. Two sets of western blot samples are shown. (c-d) Image analysis of MBP protein expression. (c) Total MBP signal (black and red arrows in (a), (b)) normalized by β -Actin

signal is shown. (d) MBP isoform (Lower band, red arrows in (a), (b)) normalized by β -Actin signal is shown.”

-Table 1 is added twice in the manuscript.

=> Thank you for your comment. This happened during the process of PDF construction of submission. This should be corrected in the publication process.